



# How changing the height of the Antarctic ice sheet affects global climate: A mid-Pliocene case study

Xiaofang Huang[1, 2*], Shiling Yang[1, 2, 3*], Alan Haywood[4], Julia Tindall[4], Dabang Jiang[5, 3], Yongda Wang[1, 2, 3], Minmin Sun[1, 2, 3], Shihao Zhang[1, 2, 3]

[1]Key Laboratory of Cenozoic Geology and Environment, Institute of Geology and Geophysics, Chinese Academy of Sciences, Beijing 100029, China

[2]CAS Center for Excellence in Life and Paleoenvironment, Beijing, 100044, China

[3]College of Earth and Planetary Sciences, University of Chinese Academy of

10      Sciences, Beijing 100049, China

[4]School of Earth and Environment, University of Leeds, Leeds, LS2 9JT, UK

[5]Institute of Atmospheric Physics, Chinese Academy of Sciences, Beijing 100029, China

15      *Correspondence to: S. Yang, yangsl@mail.iggcas.ac.cn

                    X. Huang, hxf@mail.iggcas.ac.cn



**Abstract:** Warming-induced topographic changes of the East Antarctic Ice Sheet (EAIS) could have significant influence on the climate. However, how large changes in the EAIS height could theoretically affect global climate have yet to be studied. Here, the influence of possible height changes of the EAIS on climate is investigated through numerical climate modeling, using the Pliocene as a test case. As expected, the investigation reveals that the reduction of ice sheet height leads to a warmer and wetter East Antarctica. However, unintuitively, both the surface air temperature and the sea surface temperature decrease over the rest of the globe. These temperature changes result from the higher air pressure over Antarctica and the corresponding lower air pressure over extra-Antarctic regions with the reduction of EAIS height. This topography effect is further confirmed by energy balance analyses. These findings could provide insights into future climate change caused by warming-induced height reduction of the Antarctic ice sheet.

**Keywords:** mid-Pliocene warm period; Antarctic ice sheet; height changes; sensitivity experiments

## 1 Introduction

The Antarctic Ice Sheet (AIS) is the largest component (by volume) of Earth's cryosphere (Gasson and Keisling, 2020). It accounts for almost 70% of the world's freshwater, representing a potential sea-level rise of 56.6 m (Shum et al., 2008). Its evolution has received considerable attention in climate research, as it determines the surface mass balance that has a major impact on both regional and global climate (DeConto et al., 2007; Bintanja et al., 2013; Colleoni et al., 2018; Golledge et al., 2019; Tewari et al., 2021a). The size of the present-day AIS is known to impinge substantially on synoptic and planetary scale atmospheric flow (Parish and Bromwich, 2007; Schmittner et al., 2011; Hakuba et al., 2012; Goldner et al., 2013; Grazioli et al., 2017), and the warming-induced topographic changes of the AIS in turn have significant influence on the climate (Orr et al., 2008; Tewari et al., 2021a, b). However, the effect



of the AIS height changes on future predictions of climate is still uncertain. One method of investigating this effect in a warmer-than-modern climate is to look back at past warm periods of Earth history, for example the Pliocene.

The mid-Pliocene warm period (~3.3–3.0 Ma) is the most recent period of relatively warm and stable climate in Earth's history, during which atmospheric $CO_2$ concentrations were approximately 400 ppmv (Pagani et al., 2010; Lunt et al., 2012a; Yang et al., 2018; De La Vega et al., 2020; Huang et al., 2021) and models suggested that global mean annual temperature was 1.7–5.2 °C warmer than today (Haywood et

al., 2020). This period is similar to today in terms of the continent–ocean configuration and atmospheric $CO_2$ concentrations (Haywood et al., 2016) and has often been proposed as a climatic analog for the end of this century (Burke et al., 2018). The present atmospheric $CO_2$ concentration is over 410 ppmv and has reached the Pliocene level. However, due to the large thermal inertia of the oceans (Levitus et al., 2000; Back

et al., 2013), the global mean temperature is projected to rise to the level of the Pliocene as early as the 2040s (Zhang, 2012; Ding et al., 2014; Jiang et al., 2016; Burke et al., 2018; Tierney et al., 2020). Therefore, we use the Pliocene as a test case to investigate how large changes in the East AIS (EAIS) height affect the climate.

        Numerical experiments have emerged as an efficient means of understanding past

climates on regional and global scales (Huang et al., 2019). Based on simulations, the dynamic behavior of the AIS and its stability to the climate change have been analyzed (Raymo et al., 2006; Naish et al., 2009; Cook et al., 2013; Patterson et al., 2014; Austermann et al., 2015; Boer et al., 2015; Yamane et al., 2015; Scherer et al., 2016; Dolan et al., 2018). Here we design sensitivity experiments using a coupled climate

model to investigate how perturbations in the EAIS heigh would interact with the atmospheric flow and influence the temperature and precipitation dynamics over the region and the rest of the planet.

## 2 Methods

### 2.1 Model description



The Hadley Centre coupled climate model version 3 (hereafter referred to as HadCM3) was used for this study. This model has been used extensively for studies of the Pliocene within the Pliocene Model Intercomparison Project experiments (Haywood et al., 2010, 2011; Bragg et al., 2012; Hunter et al., 2019). HadCM3 consists of two main components: an atmospheric component (HadAM3) and an oceanic component (HadOM3) (Gordon et al., 2000; Pope et al., 2000; Valdes et al., 2017). The horizontal resolution of the atmosphere model is 2.5° in latitude by 3.75° in longitude and consists of 19 layers in the vertical. The atmospheric model has a time step of 30 min and includes a radiation scheme that can represent the effects of major and minor trace gases (Edwards and Slingo, 1996). The HadOM3 spatial resolution over the ocean is 1.25° latitude by 1.25° longitude, with 20 vertical layers. The ocean model is a 'rigid lid' model, which has a time step of one hour and incorporates a thermodynamic-dynamic sea ice model with primitive (ocean drift) dynamics. The HadCM3 has been shown to well represent the broad-scale features of the Antarctic and Arctic atmospheric and oceanic circulation (Turner et al., 2006; Chapman and Walsh, 2007). The fact that the HadCM3 consistently performs well in tests against other coupled atmosphere–ocean models (Lambert and Boer, 2001; Hegerl et al., 2007; Dolan et al., 2011) increases our confidence in its palaeoclimate simulations.

## 2.2 Pliocene boundary conditions and experimental designs

For this study the required mid-Pliocene boundary conditions were supplied by the U.S. Geological Survey Pliocene Research Interpretations and Synoptic Mapping Group's (PRISM) dataset, specifically the latest iteration of the reconstruction known as PRISM4 (Dowsett et al., 2016). They include topography and bathymetry, coastlines, land surface properties (i.e., vegetation, soil type, and ice sheet coverage) and atmospheric composition with respect to pre-industrial conditions. The Greenland Ice Sheet and the West Antarctic Ice Sheet, which currently store ~13 m sea-level equivalent ice (Dolan et al., 2011; Yamane et al., 2015), are thought to have largely melted during the mid-Pliocene warm period (Lunt et al., 2008; Naish et al., 2009).



Therefore, our experiments focus on changing the East Antarctic Ice Sheet height. It
should be noted that the surface type is still 'snow' and so there will still be high albedo
in this region.

Our simulations are started from the end of the HadCM3 contribution to PlioMIP2
simulation (Hunter et al., 2019). There are two differences between our simulations and
the PlioMIP2 simulation: i) we use dynamic vegetation (Hunter et al. (2019) uses fixed
vegetation from PRISM4); ii) The height of the Antarctic ice sheet is constant in
PlioMIP2 simulation (Hunter et al., 2019), but is changed successively in our study. To
evaluate the regional and global climate sensitivity to the EAIS height changes, five
Pliocene modelling experiments are presented in this paper, which were identical except
for the height of the EAIS: one mid-Pliocene control run (hereafter MPControl) and
four sensitivity simulations with height reduced by 100% (hereafter 0%EAIS), 75%
(hereafter 25%EAIS), 50% (hereafter 50%EAIS), and 25% (hereafter 75%EAIS) of the
Pliocene height.

The first experiment, which we term Pliocene control, uses the East Antarctic ice
sheet configuration (and all other boundary conditions) specified in the USGS PRISM4
data set. All experiments (including the ice sheet sensitivity experiments) are started
from the end of the HadCM3 PlioMIP2 simulation and are continued for another 500
model years allowing the modelled climate to be equilibrated to the boundary
conditions. Climate statistics are based on time averages of the final 30 years for each
run. The results are presented as anomalies from the control for the sensitivity
experiments, thereby estimating the EAIS height effect during the mid-Pliocene warm
period.

### 3 Results

**3.1 Temperature changes**

Reducing the height of the EAIS experiments results in a dramatic annual mean
warming over East Antarctica relative to the MPControl experiment (Figure 1).
Compared with the MPControl experiment, the East Antarctic annual surface



temperature increased by about 5 °C, 10 °C, 15 °C, and 18 °C with the height reduction

of 25%, 50%, 75%, and 100%, respectively (Figure 1). This surface warming, occurring

at a rate of approximately 5 °C per kilometer of EAIS height lost, is accompanied by a

prominent surface cooling over western Antarctica and the Southern Ocean.

Contrary to Antarctic warming, reducing the height of the EAIS experiments leads

to annual mean surface cooling over the rest of the globe (Figure 2). The inclusion of

the 0%EAIS set of boundary conditions results in a ~1–2 °C mean cooling over the rest

of the globe (Figure 2a). In low and equatorial regions, temperatures decrease by a

minimum of 0.5–1 °C and cooling is at its greatest (~3 °C) over Southern Ocean. For

25%EAIS and 50%EAIS experiments (Figures 2b, c), annual mean values for surface

air temperature decrease by ~0.5 °C and ~1 °C, respectively. Compared with the

MPControl experiment, the surface air temperature in 75%EAIS experiment changes

little (the mean value near zero; Figure 2d).

Analysis of sea surface temperature (SST) for all sensitive experiments shows the

presence of the cooling, which extends across all ocean basins of the world (Figure 3).

SST decreases are greatest in 0%EAIS (~1–2 °C; Figure 3a), while smallest in

75%EAIS (~0 °C; Figure 3d). Moreover, similar to the anomalous patterns of the SAT,

the global surface ocean is — with a few exceptions of regional warming —

characterized by decreased SST, a pattern that is more pronounced in the Southern

Ocean.

### 3.2 Precipitation changes


The numerical simulations show that with the height reduction of the EAIS, the

annual precipitation has increased over East Antarctica (Figure 4). Precipitation

enhancements are greatest in 0%EAIS (~0.4 mm day$^{-1}$; Figure 4a) and smallest in

75%EAIS (~0.1 mm day$^{-1}$; Figure 4d). This precipitation enhancement, occurring at a

rate of approximately 5% per degree Celsius of temperature, is accompanied by a

precipitation deficit over the western Antarctica and the Southern Ocean. With respect

to the MPControl experiment, precipitation reduces significantly over the western





Antarctica and the Southern Ocean (~0.3–0.8 mm day$^{-1}$; Figure 4a) in the 0%EAIS experiments, but decreases slightly over those areas (~0.1–0.2 mm day$^{-1}$; Figure 4d) in

the 75%EAIS experiments.

Annual precipitation decreases consistently over most areas on the globe in all the sensitivity experiments compared to the MPControl experiments (Figure 5). This is consistent with the decreased air temperatures (Figure 2), which reduce moisture carrying capacity of the air and lead to less precipitation. The experiment showing the

greatest sensitivity in terms of precipitation response is 0%EAIS, with the anomaly varying from -2 to 0.8 mm day$^{-1}$ (Figure 5a), while the least is 75%EAIS with a narrow anomalous range of -0.4–0.4 mm day$^{-1}$ (Figure 5d). The spatial patterns (Figure 5) show that the enhanced precipitation focuses over parts of the tropics and the 45th parallel south, while the deficit focuses over northern high latitudes and the Antarctic

periphery. The largest precipitation anomaly is found in the tropics that are dominated by the intertropical convergence zone (ITCZ). In general, for most areas except the Southern Ocean, the simulations that display the largest SAT sensitivity to the prescription of EAIS height changes also exhibit the largest precipitation anomaly.

**4 Discussion**

**4.1 Wind over southern hemisphere**

Earlier studies have shown a clear relationship between the atmospheric circulation and precipitation dynamics, arguing that precipitation over polar regions is mostly due to orographic effects acting upon the circulation pattern passing over the

region (Schmittner et al., 2011; Hakuba et al., 2012; Goldner et al., 2013; Tewari et al., 2021a). The mechanical obstruction by the ice sheet prevents the moisture laden winds from penetrating inland (Parish and Bromwich, 2007; Grazioli et al., 2017; Tewari et al., 2021b). The precipitation increases over EAIS under the successive topographic reduction (Figure 2), which is causally related to the elevated moisture transport into

the continent due to the weakened katabatic flow (Goldner et al., 2013; Tewari et al., 2021b).



Figure 6 shows the magnitude and direction of the low-level wind at 850 hPa over the Southern Hemisphere and the corresponding changes observed in their strength due to orographic perturbations in individual simulations. In the MPComtrol experiment, strong surface westerly winds encircle the East Antarctic continent, extending from ~30°S to the continental periphery (Figure 6a), indicating the blocking effect of the EAIS (Tewari et al., 2021b).

Upon successive reduction of the EAIS height (Figures 6b–e), the westerly flow becomes stronger between 30°S and 60°S, while it becomes weaker between 60°S and 90°S and penetrates gradually into the eastern continent. The EAIS height reductions of 100% and 75% cause a poleward shift in the surface flows (Figures 6b, c), which even circulates around the Southern Pole. In contrast, reductions by 50% and 25% cause little change in the surface winds. In this context, sustained attention needs to be paid to changes in the height of AIS in future warming and their effect on atmospheric circulation and precipitation dynamics over the region.

## 4.2 Surface air pressure

The height reduction of the EAIS causes warming over East Antarctica, which can be explained by lapse rate (Abe-Ouchi et al., 2007). This was also addressed in several studies for cases of polar ice sheets and Tibet Plateau by changing the surface elevation (Kutzbach et al., 1993; Krinner and Genthon, 1999; Abe-Ouchi et al., 2007; Goldner et al., 2013; Singh et al., 2016). However, a prominent cooling due to the EAIS reduction is observed over the rest of the globe (Figure 2). This can be well explained by the surface air pressure changes (Figure 7).

As shown in Figure 7, the surface air pressure increases over Antarctica and decreases over elsewhere, which is similar to the spatial pattern of the air temperature changes (Figure 2). With the reduction of the EAIS height, the air mass increases over Antarctica, which at the expense of that over the rest of the globe, leading to higher air pressure over Antarctica and lower one over extra-Antarctic regions (Figure 7). According to the ideal gas law (Clapeyron, 1834), lower air pressures translate to lower





air temperatures, which well explains the temperature contrast between Antarctica and extra-Antarctic regions.

### 4.3 Modelling methodological limitations

In the present study, the HadCM3 model was used to investigate the influence of the height reduction of the EAIS on temperature, precipitation, atmospheric circulation, surface air pressure, and the energy transport at the regional and global scales. The objective of these simulations was to quantify how the existence of the EAIS would affect the mid-Pliocene climate. It can be concluded from the present findings that

reduction in the EAIS height during the mid-Pliocene warm period induces warming and wetting over the East Antarctica, and the cooling over the extra-Antarctica regions. The Antarctic surface warming and costal cooling due to the height reduction of Antarctic ice sheet were also observed in the modern Antarctic height reduction sensitivity experiments using the CAM5.1 model (Tewari et al., 2021a). It should be

noted that the effect of changes in the surface albedo, sea level, and continental margins, which would undoubtedly occur with such orographic variations, have not been explicitly taken into account in the present idealized simulations. Despite these caveats, we expect that the dynamical influence of the EAIS over the Antarctic presented herein will persist even in their presence.

Another modelling limitation is that the water contained in Antarctica did not get redistributed over the ocean when we reduced the EAIS height. This is because the HadCM3 is a 'rigid lid' model, which means the sea-level is essentially fixed. To provide a more realistic 0%EAIS experiment, we perform a new experiment in which the EAIS is still at 0% but the land topography (away from Antarctica) is reduced by

60m, to artificially raise the sea level. The changes between this experiment and the MPControl experiment show that the surface air temperature and surface air pressure (Figure 8) both show a similar spatial pattern with the changes between the 0%EAIS and MPControl experiments. However, the results also show that 1) the pressure difference over the land (figure 8a) is much smaller than that in figure 7a, but there is



still a pressure difference over the ocean. 2) the temperature over the land away from Antarctica is still colder (figure 8b), although is not by as much in figure 2a. Clearly, the cooling away from Antarctica is robust, and would occur even if sea level changes were accounted for. Therefore, global temperature changes are likely to result from changes in the height of the EAIS.


## 4.4 Energy balance

In order to further identify factors controlling the air temperature changes with the height reduction of the EAIS, energy balance analyses (Heinemann et al., 2009; Lunt et al., 2012b; Hill et al., 2014) between the 0%EAIS and MPControl experiments have
been completed. This approach has been used in palaeoclimate simulations to understand the simulated temperature changes (Donnadieu et al., 2006; Murakami et al., 2008; Hill et al., 2014). The results show that the heat transport by winds from the Southern Ocean to Antarctica is the primary factor influencing the temperature changes over Antarctica (Figures 6b, 9), which is consistent with the pronounced cooling over
the Southern Ocean (Figure 3a).

The secondary factor controlling the Antarctic temperature is 'Topography+GHG'. All experiments were forced with the same trace gases, therefore the 'Topography+GHG' factor represents both the direct effect of height sheet changes on temperature (see section 4.2.2), but also some indirect effects via GHG feedbacks. One
indirect effect is that when the EAIS is reduced the atmosphere will become thicker in this region, which will lead to more greenhouse gases in the column and hence more warming. Another possible indirect effect is that the warmer atmosphere will be able to hold more water vapour. Our results are useful not only for future climate projections but also for better understanding of the growth and decay of the AIS and their
interactions with climate in geological past.

## 5 Conclusions

The sensitivity of climate to the height changes of East Antarctic ice sheet during



the mid-Pliocene warm period has been conducted using the HadCM3 model. The
results show that, due to a successive topographic reduction in the East Antarctic ice
sheet, i) the surface air temperature increases at a rate of approximately 5 ℃ per
kilometer of EAIS height lost; ii) the precipitation over EAIS increases at a rate of
approximately 5% per degree Celsius of temperature; iii) the surface air temperature
and the sea surface temperature both decreases over the rest of the globe; and iv) the
surface air pressure increases over the East Antarctica, while decreases elsewhere.
Energy balance analyses show that the topography changes of Antarctica are mainly
responsible for the temperature changes. These findings could provide insights into
future changes caused by warming-induced decay of the Antarctic ice sheet.

**Data availability**

The data presented in the figures can be downloaded from the server located at
the School of Earth and Environment of the University of Leeds. Contact Julia Tindall
(j.c.tindall@leeds.ac.uk) for access.

**Author contributions**

Xiaofang Huang contributes to the experiments, data analysis, idea and draft
paper. Shiling Yang provides the funding acquisition, and helps to revise the draft.
Alan Haywood contributes to the experiments design and helps to revise the draft.
Julia Tindall assists to perform the experiments and helps to revise the draft. Dabang
Jiang helps to revise the draft. All authors make contributions to the paper discussion.

**Competing interests**

The authors declare that they have no conflict of interest

**Acknowledgements**

This study was supported by the National Natural Science Foundation of China
(41725010 and 42107472), the Strategic Priority Research Program of the Chinese



Academy of Sciences (XDB26000000 and XDB31000000) and the Key Research Program of the Institute of Geology & Geophysics, CAS (IGGCAS-201905).

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

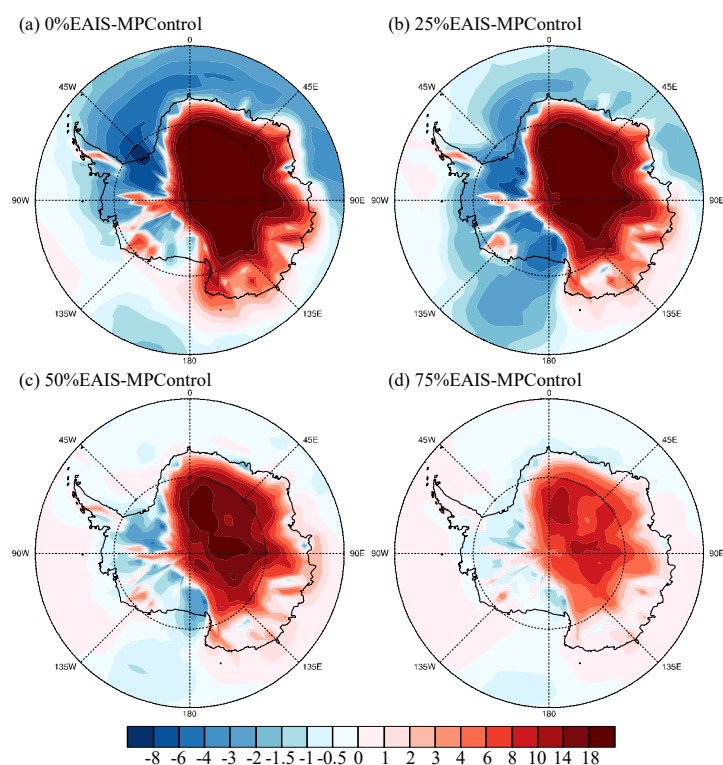

Figure 1. Spatial distribution of the annual mean surface temperature anomalies

(units: ℃) over Antarctica between sensitivity experiments and MPControl

experiments.



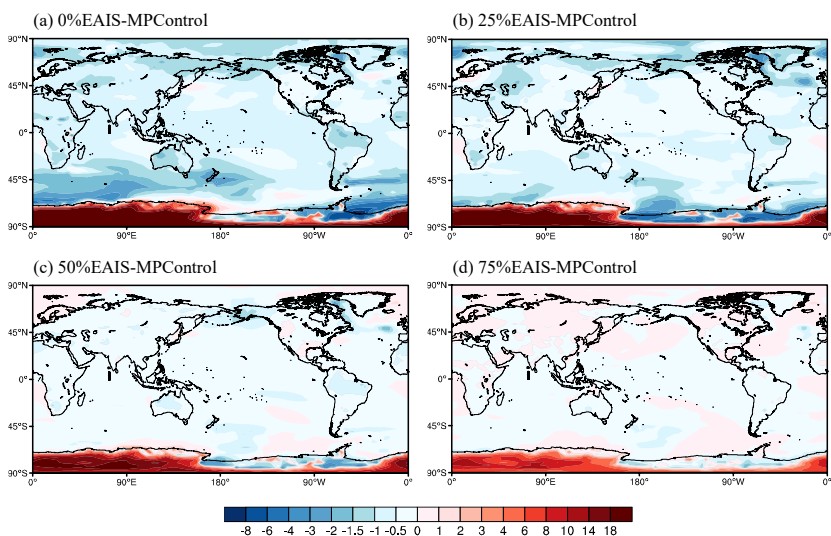

Figure 2. Spatial distribution of the annual mean surface air temperature anomalies
(units: °C) over the globe between sensitivity experiments and MPControl
experiments.






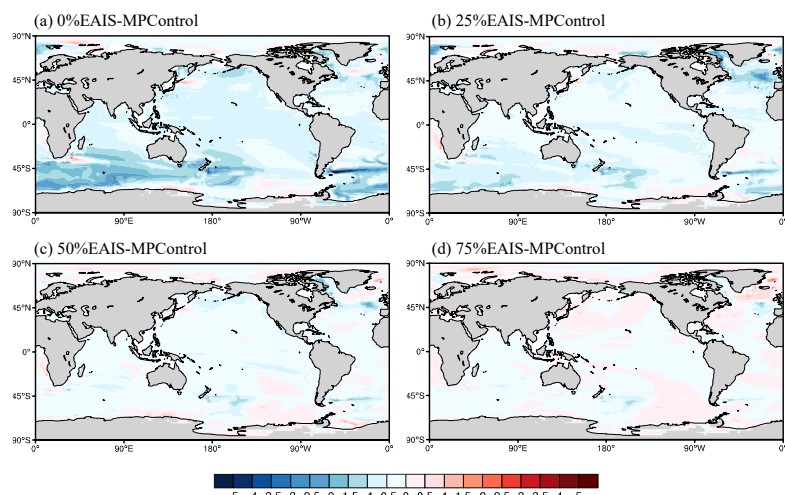

Figure 3. Spatial distribution of the annual mean sea surface temperature anomalies

(units: °C) over global between sensitivity experiments and MPControl experiments.

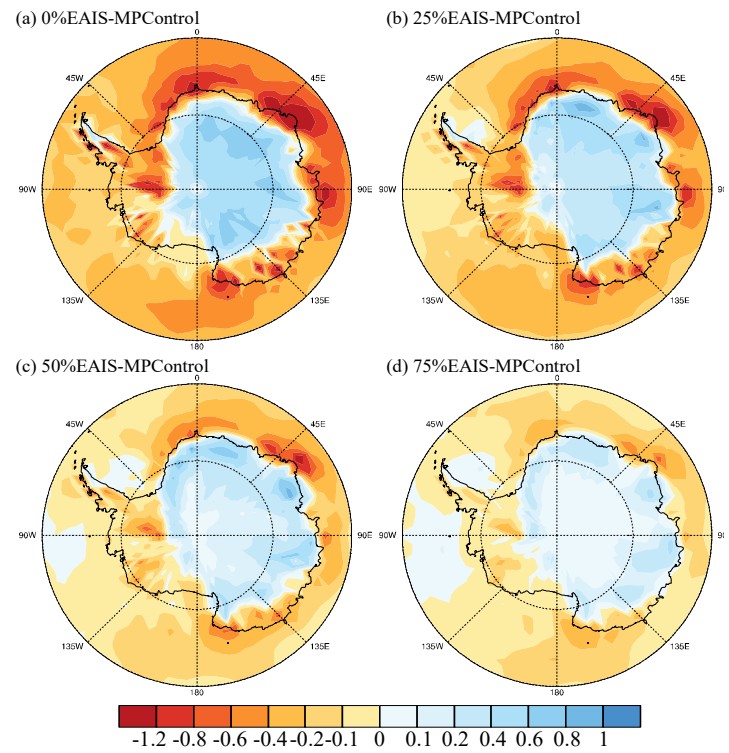

Figure 4. Spatial distribution of the annual mean precipitation anomalies (units: mm day$^{-1}$) over Antarctica between sensitivity experiments and MPControl experiments.


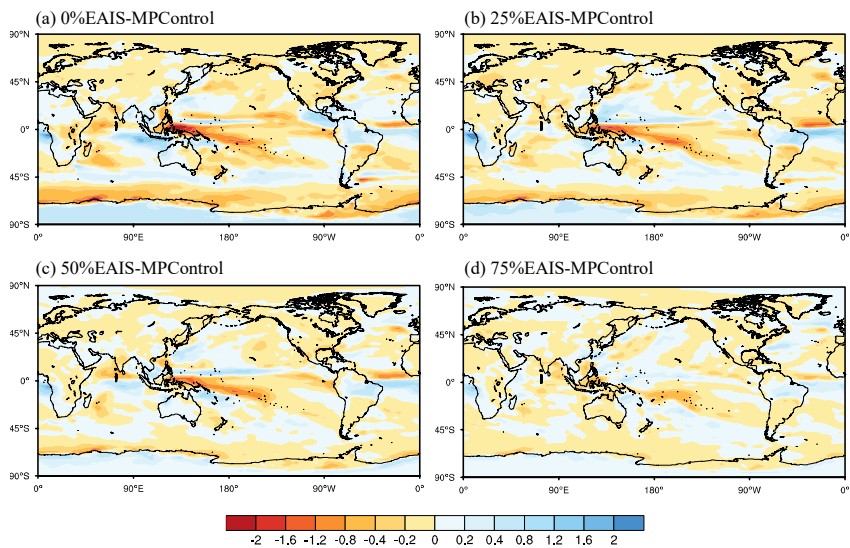

Figure 5. Spatial distribution of the annual mean precipitation anomalies (units: mm day$^{-1}$) between sensitivity experiments and MPControl experiments.





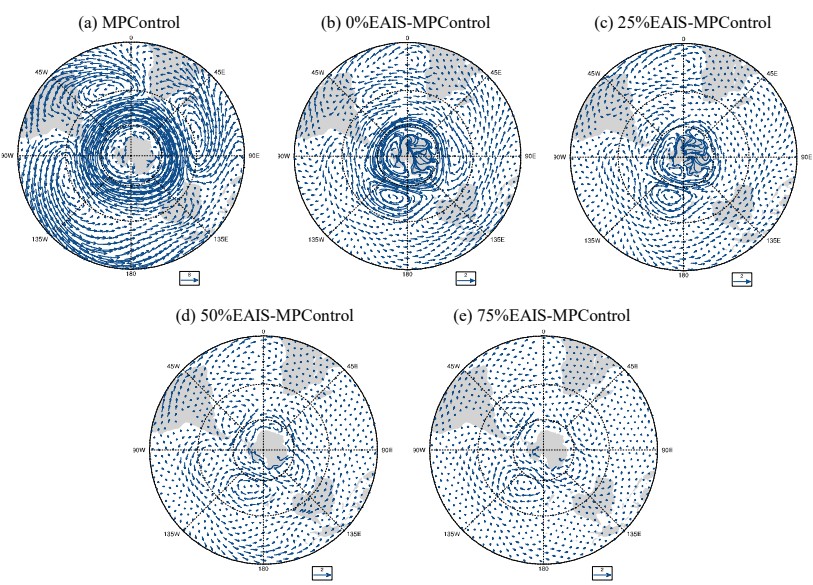

Figure 6. Annual mean wind circulation at 850 hPa over the Southern Hemisphere (a;

units: m s$^{-1}$) and its corresponding anomalies in 0%EAIS, 25%EAIS, 50%EAIS, and

75%EAIS, respectively (b-e; units: m s$^{-1}$).



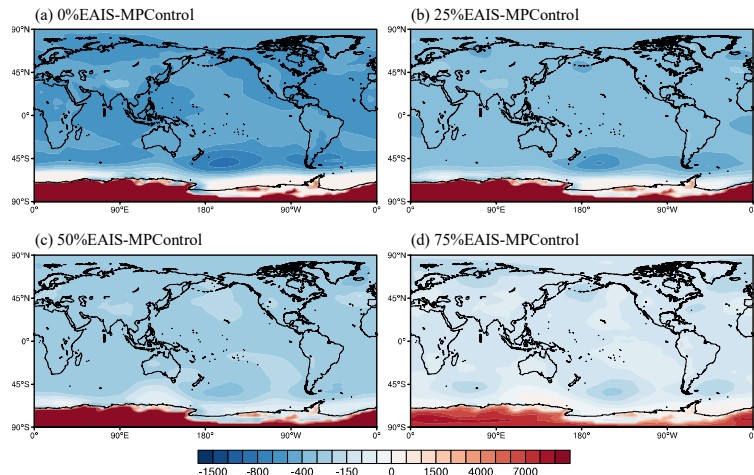


Figure 7. Spatial distribution of the annual mean surface air pressure anomalies (units:

Pa) between sensitivity experiments and MPControl experiment.





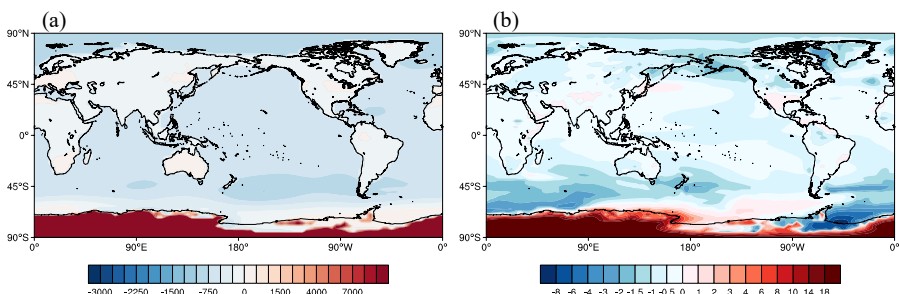

Figure 8. Spatial distribution of (a) the annual mean surface air pressure anomalies (units: Pa) and (b) the annual mean surface air temperature (units: °C) between the new sensitivity experiment and MPControl experiment. The new sensitivity experiment is similar to the 0%EAIS experiment, except artificially raising the sea level by reducing the land level (away from Antarctica) by 60m.



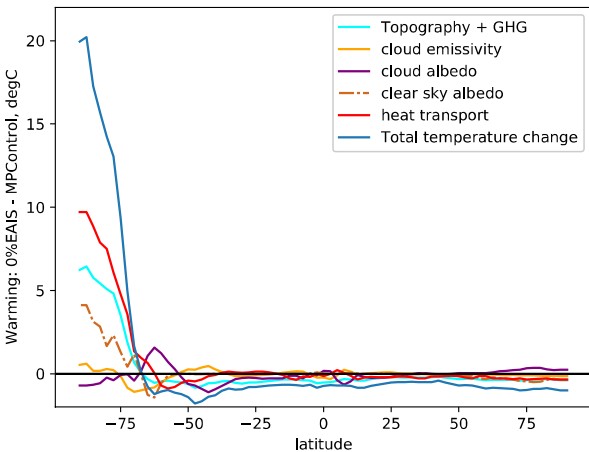

Figure 9. Energy balance analysis between 0%EAIS and MPControl. Plot shows the
zonal mean warming/cooling at each latitude, from each of the energy balance
components. GHG stands for greenhouse gases.
