# Peer review of "How changing the height of the Antarctic ice sheet affects global climate: A mid-Pliocene case study"

_Climate of the Past, 2022_

## Author Comment (AC1)

**General comments**

Huang et al. presented an interesting analysis of the global-scale effects of changes in ice sheet volume. It is useful to see the local, regional, and global changes in climate variables caused by reductions in the height of the east Antarctic Ice Sheet. The results were clearly and systematically presented. The methodological limitations of the model were nicely discussed and addressed.

However, it is unclear to me whether this is intended as an idealized study, or whether it seeks to replicate the effects of actual past and future changes in ice sheets. If the former, that should be made clear. If the latter, then there needs to be a much more robust discussion of the relevant past and/or future scenarios that are meant to be reproduced here. If these experiments are meant to investigate possible future changes in climate, then it would also be useful to see comparisons between the Pliocene control simulation and pre-industrial control, given that the Pliocene control simulation is run with PRISM4 boundary conditions that include significantly reduced ice sheet volume (specifically over West Antarctica and Greenland) as compared to the present-day. In general, it would be useful to see a more detailed discussion of how the sensitivity experiments presented here correspond to past/future scenarios that have been studied in the scientific literature.

Yes, our simulations are intended as idealized studies. As HadCM3 is a 'rigid lid' model, the water contained in Antarctica did not get redistributed over the ocean when we reduced the EAIS height, which means the sea-level is essentially fixed. Therefore, the effect of changes in the surface albedo, sea level, and continental margins, which would undoubtedly occur with such orographic variations, have not been explicitly taken into account in our simulations (see section 4.3 Modelling methodological limitations; lines 251-258). We add some words to make the expression clear (line 65).

**Specific comments**

1)  Paragraph starting on line 50: You discuss using the mid-Pliocene warm period as an analog for end-of-century climate. The time-scales for changes in different

parts of the Earth system differ; as you discuss, although present-day $CO_2$ concentrations are similar to the Pliocene, it will take time for Earth's global mean temperature to rise to Pliocene levels. It will also take time for vegetation to adjust to the Pliocene climate, and—importantly—for ice sheet loss comparable to Pliocene conditions to occur. I think this section needs to include some discussion of the existing scientific understanding of future changes in the volume of the East Antarctic Ice sheet, including the possible time-scales of ice sheet loss. How far into the future might we expect to see a mid-Pliocene-like East Antarctic Ice sheet volume?

Done (lines 63-64). Thanks for the suggestion.

2) Line 105, "our experiments focus on changing the East Antarctic Ice Sheet height": this makes it sound like you are changing the East Antarctic Ice Sheet height against its modern or pre-industrial value, but you are changing the East Antarctic Ice Sheet height against its reconstructed Pliocene value.

Thanks. We have revised the sentence to make it clear (see line 109).

3) Paragraph starting at line 108: I'd like to see more justification for this experimental design. Are the 0%EAIS, 25%EAIS, 50%EAIS, and 75%EAIS experiments intended to represent analogs for possible future scenarios, and if so under what conditions and over what time-scales could these scenarios arise?

These sensitivity experiments are hypothetical scenarios. We add more justification for the experimental design (lines 122-127).

4) Line 112: Are changes in the ice sheet dynamically resolved in the model, or are you manipulating the height of the ice sheet for each sensitivity simulation? This is unclear here.

In our study, we manipulate the height of the ice sheet for each sensitivity simulation. We have improved the sentence to make it clear (lines 114-116).

5) Lines 112-118: Does the mid-Pliocene control experiment already have reduced EAIS volume, as specified in the PRISM4 boundary conditions? If so, it would be helpful to describe in more detail the differences between PRISM4 EAIS configuration and its present-day volume/extent.

In the mid-Pliocene control experiment, the EAIS volume was as specified in the PRISM4 boundary conditions (lines 128-130). The differences in EAIS volume between the mid-Pliocene and present-day have been added (lines 130-132). Thanks for the suggestion.

6) Lines 188-191: The winds bringing moisture over the continent are different from the katabatic winds mentioned; it would be helpful to be more explicit here about the causal relationships between weakened katabatic flow and elevated moisture transport.

Done (lines 199-203). Thanks for the suggestion.

7) Lines 203-205: This is too vague.

Thanks for this comment. As our sensitivity experiments are hypothetical scenarios, it's hard to be more specific there based on the preliminary results.

8) Line 208: In section 4.4, you present a nice analysis of the energy balance, and find that "heat transport by winds from the Southern Ocean to Antarctica is the primary factor influencing the temperature changes over Antarctica." Line 208 makes it sound like the atmospheric temperature lapse rate is the primary factor for warming over East Antarctica, which seems to contradict your findings in section 4.4.

Based on the analysis of the energy balance (Figure 9), we found that the primary factor is actually heat transport. However, the topography (which represents the lapse rate) is also important (turquoise line in Figure 9). We did not say that the atmospheric temperature lapse rate is the 'primary' factor on line 208. We are sorry

for the misleading sentence, which have been rewritten (lines 223-225). Moreover, we add some words to make the expression more clear (line 290).

9) Line 230: was EAIS height reduced below the PRISM4 reconstructed height during the mid-Pliocene warm period? PRISM4 focused on a specific interglacial period, so the height of ice sheets would have fluctuated during the mid-Pliocene. But is there evidence to suggest that the EAIS would have disappeared completely? Or are these hypothetical scenarios? Again, the justification for the experimental design needs to be more clear.

   This is the same question posed in specific comment 3. See the responses above.

10) Line 245: would this have effects on ocean gateways such as the Bering Strait, and what impact might this have on ocean dynamics? Would these effects be significant?

   Yes, that is correct. Reducing the height of the land could open up some gateways that are closed in our experiments. However, this experiment was designed to remove the unrealistic surface air pressure anomaly over the land (Figure 8a), and see how this affected the surface air temperature anomalies. Therefore, we add some sentences to make the experiment design more clear (lines 261-265)

11) Section 4.4: Please add more detail about how you conducted this energy balance analysis.

   Done (lines 282-283).

12) Line 273: Which of these sensitivity experiments are applicable to which future and/or past climate scenarios? Please be more specific here.

   Thank you for the suggestion. Our sensitivity experiments are hypothetical scenarios. It's hard to specify which future and/or past climate scenarios based on the preliminary results. To avoid misunderstanding, we rewrite the sentences (lines 295-297)

13) Line 278: similar to previous comments—is there evidence for these changes in EAIS height actually occurring during the mid-Pliocene warm period? Or are these hypothetical scenarios?

These are hypothetical scenarios. This is the same question posed in specific comment 3. See the responses above.

**Technical corrections**

1) Line 59: would make more sense to write: "due to the large thermal inertia of the oceans, the global mean temperature is not projected to reach the level of the Pliocene until the 2040s."

Done (line 60). Thanks for the suggestion.

2) Line 85: this is the spatial resolution of, not over, the ocean—correct?

Done (line 88).

3) Line 194: Typo, MPComtrol to MPControl

Many thanks. We are sorry for this mistake and have revised it (line 209).

4) Line 209: could change to "which can be explained by the lapse rate"

We have rewritten this sentence (lines 223-225).

5) Line 219: rewrite as "leading to higher air pressure over Antarctica and lower air pressure over extra-Antarctic regions."

Done (line 235).

6) Line 220: perhaps it would make more sense to replace "translate to" by "correspond with."

Done (lines 236-237). Thanks for the suggestion.

7) Line 232: "costal" to "coastal"

Done (line 248).

8) Line 269: there is no Section 4.4.4

Many thanks. We are sorry for this mistake and have revised it (line 290).

9) Line 284-285: should be "the surface air temperature and the sea surface temperature both decrease…. The surface air pressure increases over East Antarctica, while decreasing elsewhere"

Done (line 307). Thanks for the suggestion.

10) Line 286: awkward sentence. Could rewrite as: "Energy balance analyses show that the temperature changes over Antarctica are mainly caused by topographic changes in the EAIS."

Done (lines 308-311). Thanks for the suggestion.

---

## Author Comment (AC3)

The authors present a model study on the impact of a reduced East Antarctic Ice Sheet in context of the mid-Pliocene Warm Period, using the HadCM3 model which took part in the PlioMIP2. They present the results of a number of sensitivity studies in which the height of the EAIS is gradually reduced. This study is relevant and the results are interesting, but I am missing a thorough scientific basis beneath many of the results presented. The study is also missing a good discussion regarding how these results can be interpreted in light of the present/future climate, as they are based on Pliocene simulations. The latter were shown several times to be highly dependent on the mid-Pliocene boundary conditions, stressing on the importance of a good assessment of the state-dependency of the system. I believe a considerable improvement can be made in order to better explain the results and provide more context on how they can be interpreted in light of present-day climate.

**General comments:**

1) Terminology regarding EAIS is a bit confusing; '0% EAIS' for the largest anomaly is not very intuitive. Consider adjusting to e.g. -25/-50/-75/-100%, and 0% or 'original' for the default configuration.

We have changed the names of the sensitivity experiments following the suggestion (lines 117-118), except for the "0% or original" for the default configuration. This is because the "0% or original" is a control experiment. Thus we prefer to name it as "MPControl", following conventional use.

2) Does the inclusion of dynamic vegetation have any significant impact compared to the original configuration?

In this study, all the boundary conditions (including the vegetation) are the same except for the height of the East Antarctic Ice Sheet. Therefore, it is difficult to discuss the effect of the dynamic vegetation just based on our sensitivity experiments.

3) Overall, figures of different experiments are rather repetitive. It could be more informative to show e.g. anomalies normalized by the 0% EAIS anomaly, to check

whether the other experiments result mostly in a linear response of the strongest signal.

Our study has already analyzed the anomalies normalized by the MPControl experiment (100%EAIS instead of 0%EAIS). The results show a linear response of temperature and precipitation to the EAIS height changes, i.e. a warming of 5 ℃ per kilometer of EAIS height lost and a precipitation enhancement of ~5% per ℃ (lines 140-143, 166-168). We believe that the two approaches (100%EAIS and 0%EAIS) answer the same question.

4) The paper is quite descriptive, I am missing a more mechanical insight into the responses shown. Many of the claims or explanations are not supported by what figures show, or not shown at all, making it hard to follow the discussion of the results.

We have improved the Figures 6 and 10 following the suggestions of reviewer #2 (see Figures 6, 10 in the manuscript). In addition, a new Figure (Figure 1 in the manuscript) has been added in our paper and some sentences have been added to make the claims or explanations more clear (lines 163-164, 196-200).

5) Subsections 4.1 and 4.2 seem to be mostly results and should therefore at least partly move to section 3?

Thanks for the suggestions. Subsections 4.1 and 4.2 aim to explain the changes in Antarctic precipitation and global temperature, respectively. Therefore, we would rather keep them in the Discussion section. To make the expression more clear, we have replaced the titles of Subsection 4.1 and Subsection 4.2 with "Cause of Antarctic precipitation changes" (lines 188-189) and "Cause of global temperature changes" (line 219), respectively.

6) Structure can be improved; many of the analyses implemented are presented 'on the go', rather than in the methods section up front along with their motivation. This would make the overall storyline clearer.

Done. We have added some sentences in the methods section to make it clearer (lines 119-124).

**Specific remarks:**

1) L19: surely there are studies? e.g. work of DeConto et al, Gasson et al.

DeConto et al. (2016) investigated the contribution of Antarctic ice sheets to past and future sea-level rise, and Gasson et al. (2016a, b) evaluated the climate effect of Antarctic ice sheet changes in the Miocene. Our sensitivity experiments are hypothetical scenarios, which focus on the EAIS height changes and their climate effect during the mid-Pliocene. We have added some words to make the expression more clear (line 20).

2) L25: temperature changes as a result of pressure changes: how are these linked?

As shown in Figure 8 in the manuscript, the surface air pressure increases over Antarctica and decreases elsewhere, which is similar to the spatial pattern of the air temperature changes (Figure 3 in the manuscript). With the reduction of the EAIS height, the air mass increases over Antarctica at the expense of those over the rest of the globe, leading to higher air pressure over Antarctica and lower air pressure over extra-Antarctic regions (Figure 8 in the manuscript). According to the ideal gas law (Clapeyron, 1834), lower air pressures correspond with lower air temperatures, which well explains the temperature contrast between Antarctica and extra-Antarctic regions.

3) L136: 5C/km is much lower compared to free tropospheric lapse rate (usually ~7K/km, often ~8C/km over ice sheets), is there an explanation for this?

A previous study has shown that the lapse rate over the Greenland ice sheet depends strongly on background climate (Erokhina et al., 2017). Specifically, the lapse rates for the early Holocene, preindustrial and observational periods are within the range of ~5.5 and 9.5°C km$^{-1}$, while the LGM lapse rates are up to 4°C km$^{-1}$ higher than the interglacial values. Therefore, we believe that the low lapse rate obtained in

our study (5°C/km) may result from the warm conditions in the mid-Pliocene, which is consistent with the finding of Erokhina et al. (2017).

4) L167: Some decrease in precipitation can indeed be expected at lower temperatures, but can you also estimate how much? Does that explain the changes seen? Apart from the global precipitation reduction outside of Antarctica, I hardly see any correlation between the temperature and precipitation anomaly patterns, so clearly other processes are at play to explain the regional responses.

Our results show that annual precipitation decreases consistently over most areas on the globe in all the sensitivity experiments compared to the MPControl experiments. This is consistent with the decreased air temperatures, which reduce moisture carrying capacity of the air and lead to less precipitation (lines 173-176). However, precipitation varies from region to region, it is hard to estimate how much it has been decreased.

It's true that, except the global pattern, the correlation between the temperature and precipitation anomaly patterns is hard to see. Our results show that he largest precipitation anomaly is found in the tropics that are dominated by the intertropical convergence zone (ITCZ). In general, for most areas except the Southern Ocean, the simulations that display the largest SAT sensitivity to the prescription of EAIS height changes also exhibit the largest precipitation anomaly (lines 183-185). The temperature changes may lead to the southward shift of the ITCZ, which contribute to the regional precipitation changes.

5) L175: The precipitation response seems to occur mostly in the South Pacific ITCZ and SPCZ, can you explain why?

The ITCZ is a zone of convergence at the thermal equator where the trade winds meet. It is a narrow band of intense precipitation and migrates with the changing position of the thermal equator. Based on our results, the temperature increases over Antarctica with the successive reduction of the EAIS height, which may lead to the

southward shift of the ITCZ. This reasonably explains why the precipitation response occurs mostly in the South Pacific ITCZ and SPCZ.

6) L188: It would be very helpful here to make a simple budget analysis of the zonally averaged southward moisture transport at different atmospheric levels. The strongest precipitation responses extend quite far over the ocean, suggesting that reduced baroclinicity may play an important role as well.

Thanks for the suggestion. As shown in Figure 7 in the manuscript, the weakened katabatic flow, due to the successive topographic reduction, leads to an elevated moisture transport into the continent, which well explains the increased precipitation over EAIS (Figure 5 in the manuscript; lines 196-200). Anyway, it is worthy of further study on the moisture transport at different atmospheric levels, as well as on the changes in baroclinicity, which will definitely be included in our future work.

7) L198: again, it would be nice to know whether the responses of the different experiments are linearly related to the EAIS reduction factor and if not how they can be explained.

Yes, the responses of the different experiments are linearly related to the EAIS reduction factor (see answers to the general question 3).

8) L220: I doubt whether this seemingly very simple reasoning explains what is going on; besides the global pattern the temperature and pressure responses do not seem to be that well correlated either. What about circulation changes, heat transports, radiative effects?

In our study, we focus on the temperature and pressure contrast between Antarctica and extra-Antarctic regions. The results show that the temperature and pressure both increase over Antarctica and decrease over extra-Antarctic regions. We analyzed the energy balance (Figure 10 in the munascript) which represents a combined result of heat transport, topography, GHG, cloud, and albedo. We found that heat transport is the primary factor influencing temperature, and the topography (which represents the ideal

gas law) and GHG play a secondary role (turquoise line in Figure 10). As for the discrepancies in temperature and pressure responses over relatively small scales, the internal feedback should be important, which requires further study.

9) If it is purely the effect of pressure, you should use the ideal gas law and estimate the temperature response from the pressure response and compare it to the actual temperature change found.

Based on the analysis of the energy balance (Figure 10 in the manuscript), we found that the primary factor is actually heat transport. However, the topography is also important (turquoise line in Figure 10). We did not say that the temperature changes are purely the effect of pressure changes. We are sorry for the misleading sentence, which have been rewritten (lines 220-221). Moreover, we add some words to make the expression more clear (line 234).

10) L234: Your abstract suggests that such studies do not yet exist?

Sorry for the misleading expression. Tewari et al., (2021; L234) addressed future climate changes, while we focus on the studies of Pliocene warm period.

11) L238: Can you support this statement?

I think this statement is a reasonable inference. To support this statement, we performed a new experiment in which the EAIS height has been reduced 100% but the land topography (away from Antarctica) is reduced by 60m, to artificially raise the sea level (lines 255-258). The results show that the cooling away from Antarctica is robust, and would occur even if sea level changes were accounted for.

12) L260: This EBM approach was also used for the Eocene by Lunt et al 2021 and for the Pliocene by Baatsen et al 2022.

The references have been added (line 275).

13) L262: The heat transport component indeed seems to be quite important over Antarctica. I do not follow how a cooler Southern Ocean is linked to higher Antarctic temperatures here? Also, it would be very useful to separate the temperature gradient and circulation components of the meridional heat transport.

Based on the energy balance analysis (Figure 10 in the manuscript), the heat transport component is the primary factor influencing the temperature changes over Southern Ocean. The heat transport increases significantly over Antarctica, while it decreases over the rest of the globe. We think the heat transport from the rest of the globe, especially from the Southern Ocean, to Antarctica is the primary factor influencing the temperature changes over Antarctica (Figures 10). This may result from the proximity of the Southern Ocean to Antarctica. We have rewritten the sentence to make the expression more clear (lines 276-279).

Thanks for the suggestion on separating the temperature gradient and circulation components of the meridional heat transport. We would conduct such analysis in our future work.

14) L281: I do not find this number anywhere in the results, how was it determined? Same for the 5% precipitation increase per degree C.

Both the temperature and precipitation numbers are shown in the results section (lines 140-143, 166-168).

15) L286: Yet, you show that the heat transport is more important in the EBM analysis?

Yes, based on the analysis of the energy balance (Figure 10 in the manuscript), we found that the heat transport is the primary factor influencing the temperature changes, which ultimately result from the topography changes of Antarctica. To make the expression more clear, we have rewritten this sentence (lines 300-301).

16) L287: This seems to be more of a motivation, rather than a conclusion from the results.

Thanks. We have deleted the sentence (lines 302-303).

**Figures:**

1) Missing a figure showing the heights and/or height anomalies applied in the experiments.

   Done. The figure has been added (Figure 1 in the manuscript).

2) Figure 2: it would be helpful to remove the idealised lapse rate effect due to elevation changes, to distinguish with other dynamical/feedback effects.

   The lapse rate is actually obtained from Figure 1 rather than Figure 2. We would like to keep it just for reference.

3) Figure 3: SST responses are almost identical to SAT responses, so I'm not sure what this figure adds besides using a more practical colour scale. Maybe showing the full-depth or upper x meter average temperature response would reveal some more fundamental circulation-related impacts. In fact, I am missing any ocean circulation responses in the figures shown.

   Yes, the SST responses are almost identical to SAT responses. I think this means that the height reduction of the EAIS has similar effect on SAT and SST. We have tried our best to adjust the colour scales of Figure 2 and 3. As the SST is an efficient indicator for ocean temperature and is widely used for analyzing patterns of climate variability. We think that the SAT parameter is sufficient to address the effect of EAIS height changes, and would keep the suggestion in mind in our future investigations.

4) Figure 4: again this figure is rather repetitive between the experiments. While this is useful to know, it does not give any explanation of the patterns seen. Are these the direct result of elevation changes, or rather e.g. the related temperature/circulation changes? What are the seasonal responses?

   Figure 4 in the manuscript shows the spatial distribution of the annual mean precipitation anomalies over Southern Hemisphere between each sensitivity experiment and MPControl experiment. The explanation of the patterns is given in Subsection 4.1

(lines 190-217), because it is not appropriate to discuss it in the results section. Our experiments were deigned to investigate the effect of EAIS height changes, and we believe the precipitation patterns are the result of elevation-induced changes. As all figures show annual results, we thus present the precipitation annually instead of seasonally to be consistent and facilitate comparison.

5) Figure 5: precipitation anomaly plots are always challenging to interpret, as there is already substantial variability in the reference, without which it is tough to see what is relevant.

   Done. We have added the reference plot into Figure 5 (Figure 6e in the manuscript).

6) Figure 6: This is a very useful figure, but hard to read. Why show the entire Southern Hemisphere, rather than e.g. 30S-90S? The projection used seems to be cylindrical, which contracts Antarctica at the expense of lower latitudes. Using a polar stereographic projection seems to be a more logical and practical choice here. Interpreting anomaly quiver plots is pretty challenging. I think it would help to add colour shading showing whether the anomaly induces a weakening (e.g. blue) or strengthening (e.g.) red of the flow in the MPcontrol.

   The temperature and precipitation changes both show the entire Southern Hemisphere and use the cylindrical projection. To be consistent, here we also show the entire Southern Hemisphere rather than 30S-90S and keep the cylindrical projection.

7) Figure 8: global sea level is adjusted by lowering the land by 60m, but coastlines seem unaffected? This figure also shows that besides the EAIS, temperature and pressure anomaly patterns do not correlate well.

   This experiment was designed to remove the unrealistic surface air pressure anomaly over the land (Figure 8a in the manuscript), and see how this affected the surface air temperature anomalies. Therefore, locations where the land was below 60 m are set to 0 m to maintain the mid-Pliocene land sea mask. We have added some sentences to make the experiment design more clear (lines 258-262).

8) Figure 9; it is hard to see what is going on besides Antarctica and for the largest terms. Consider changing the scaling or separating some of the components. The different components do not show actual warming/cooling, but their estimated (linear) temperature contribution from the EBM.

Done (see Figure 10 in the munascript).

**Typos/small errors:**

1) L134: increases?

Yes, it should be "increases" (line 139). We have corrected this mistake.

2) L194: MPcomtrol

Done. We are sorry for this mistake and have revised it (line 207).

3) L195: the Antarctic continent?

Our MPControl experiment uses the PRISM4 boundary conditions without any changes. As the West Antarctic Ice Sheet has been melted in the PRISM4 boundary conditions, here we use the East Antarctic continent.

4) L209: explained by lapse rate: something is missing here

To make the expression more clear, we have rewritten this sentence (lines 220-221).

5) L268: height sheet

Done. We have revised it (line 286).

6) L284: decrease

Done. We have replaced "decreases" with "decrease" (line 302).

**References**

Clapeyron, É.: Mémoire sur la puissance motrice de la chaleur. Journal de l'École polytechnique, 14, 153–190, 1834.

DeConto, R. M., and Pollard, D.: Contribution of Antarctica to past and future sea-level

rise. Nature, 531(7596), 591–597, 2016.

Erokhina, O., Rogozhina, I., Prange, M., Bakker, P., Bernales, J., Paul, A., and Schulz, M.: Dependence of slope lapse rate over the Greenland ice sheet on background climate, J. Glaciol., 63(239), 568–572, 2017.

Gasson, E., DeConto, R. M., Pollard, D., and Levy, R. H.: Dynamic Antarctic ice sheet during the early to mid-Miocene, P. Natl. Acad. Sci. USA, 113(13), 3459–3464, 2016a.

Gasson, E., DeConto, R. M., and Pollard, D.: Modeling the oxygen isotope composition of the Antarctic ice sheet and its significance to Pliocene sea level, Geology, 44(10), 827–830, 2016b.

Tewari, K., Mishra, S. K., Dewan, A., Dogra, G., and Ozawa, H.: Influence of the height of Antarctic ice sheet on its climate, Polar Sci., 28, 100642, doi:10.1016/j.polar.2021.100642, 2021.

---

## Referee Report (RR1)

**Reviewer comment on 'How changing the height of the Antarctic ice sheet affects global climate: A mid- Pliocene case study' by Huang et al**

The authors have made several useful adjustments and additions to the manuscript to address some of the issues pointed out. In contrast to what was suggested by the editor, the revisions are mostly minor and hardly any additional analysis and/or discussion was provided where requested. Therefore, a solid basis explaining the relevance of the study is still missing, as well as the elements needed for a proper physical understanding of the results shown.

In their responses, the authors seem to evade most of the issues raised and simply repeat what was already in the text, or state some trivial explanations that are often not related to the questions asked.

Many of the results and discussion focus on the obvious result, which is a lapse-rate induced temperature change and global temperature/pressure/precipitation response by redistribution of mass. The more interesting and far less intuitive responses beyond this first order effect are in my opinion left mostly untouched. I feel a more substantial effort can and should be made before publication.

Main comments:
- L112: you mention specifically that you use dynamic vegetation here, yet you answer that all the boundary conditions are the same except for the EAIS height. I assume between your simulations? This still means that there is a difference between your MPcontrol and the original PlioMIP simulation. Please clarify.
- Temperature and precipitation responses outside of the EAIS are clearly not all linear between the different experiments. Are the linear responses you mention globally averaged, or over the EAIS only? Especially with the larger reductions, both temperature and precipitation patterns become interesting and are likely related to circulation changes in both the atmosphere and ocean. These patterns as well as their dependency to the EAIS height are not fully explained.
- Improvements made to the figures are very minor and most of the issues regarding readability as well as relevance to the results and conclusions remain.
- Regardless of the changes made, section 4 is a mix of discussion and model results making this part confusing and messy. None of the analyses are presented in the methods section, and new results now seem to appear out of the blue past the main results section. The minor additions made to the methods section currently fail to resolve the overall unclear structure.

Specific comments:
- Correlation does not imply causation. Indeed, the redistribution of air causes global changes in pressure when changing the height of the EAIS. Using the ideal gas law, one can argue that warmer air will increase surface pressure. In reality, thermal heat lows would claim the opposite relation, as the atmospheric circulation also responds to the density anomalies. This is probably just a poor example in comparison to the study, but explaining the temperature response purely from the ideal gas law at least needs some more explaining. You could at least check whether the temperature and pressure anomalies and their spatial patterns are consistent with your hypothesis.

- The lapse rates found as a result of changing the EAIS height should be explained better in the text as well. I am also not sure whether this calculation is correct; in the 50% reduction case there is a >18C temperature increase over the highest region of the ice sheet, which is reduced by about 2km in elevation. This would correspond to a 9C/km lapse rate rather than 5C/km. This may be completely different from how the lapse rates were calculated, but it is impossible to tell without a proper explanation.
- The text still mentions that 'precipitation changes are consistent with decreased temperatures', without explaining any of the regional patterns, inconsistencies or non-linear responses. The -100% precipitation anomaly clearly is not twice the -50% one, for example, this is neither mentioned, nor explained. Another example is a slight precipitation increase over West Antarctica, where we see strong cooling. It is also unclear to me how the 5% precipitation increase per degree C was obtained, is this global average precipitation vs temperature?
- I am still missing any mechanical explanation as to why the ITCZ/SPCZ would respond to EAIS changes. A thermal imbalance between hemispheres can indeed be a cause (but should then be quantified) of an ITCZ shift, but it is still unclear why the effect ramps up beyond the -50% reduction.
- It is pretty much impossible to see the change in katabatic winds from figure 7. Although it may be straightforward, the wind field alone is not enough to explain moisture transports without knowing the actual moisture field. Even in the current climate, katabatic winds are confined to the area very near the ice sheet's edge, making it tough to explain moisture transports over a large latitudinal range from this effect only.
- The statement that responses to the EAIS are linear are still not substantiated by any clear figure or clear quantitative assessment. Without such, it is a claim that cannot be validated nor explained.
- If the study focuses only on the effects over Antarctica versus the rest of the globe, this is currently unclear from the abstract/introduction. As you show in the energy balance analysis, heat transports are the primary contribution to much of the changes. Yet, you claim that the ideal gas law and temperature changes are mostly responsible. This is at least partly contradictory, as the heat transport suggest that circulation changes should have at least a comparable contribution. While mentioning this contradiction yourself, the ideal gas law explanation is still presented as the main mechanism in section 4.2.
- The experiment in which the land surface is decreased by 60m does not act to support the direct link between temperature and pressure. It merely shows that the mass loss of the AIS that was previously unaccounted for does not substantially alter the results. If anything, Figure 9 shows that outside of the AIS, where lapse rate effects dominate, hardly any spatial correlation remains between the temperature and pressure responses.

Figures:
- Figure 1: this is a nice addition, but does not show any new information compared to what can be found in previous PlioMIP publications. The aim of such a figure would be to show how the EAIS was changed between the specific experiments.
- Figures 2-4: as figures 2 and 3 show the same field over a different region, while figure 4 shows a very similar field over the same region, at least one of them is redundant in the current set-up. Of course SAT and SST effects are closely related over the ocean, as they are both at or near the surface and therefore nearly the same.
- Figure 7: The projection and latitudinal extent used here is not at all consistent with Figures 2 and 5, so I fail to see what kind of consistency is meant here. Regardless of consistency, the figure remains near impossible to read and interpret. Cylindrical versus stereographic projection will not make much of a difference when looking only at the pole, but the former becomes very unrealistic when showing an entire hemisphere.

---

## Referee Report (RR2)

Referee Comment

**General comments**

Huang et al. addressed most of my comments well. However, a few of the comments were not sufficiently addressed, and there are certain points where the text remains confusing or logically inconsistent.

I think this paper could be published with minor revisions to address these issues.

**Specific comments**

Line 64, "In this scenario, Antarctica's melting ice sheets would raise sea level 20 meters in coming centuries (Grant et al., 2019).": This is not necessarily inaccurate, but it's confusing given that the aspect of AIS reduction of concern in this study is not the sea level rise but the change in the volume of the ice sheet. As you describe,  Could you make a statement here about the volume of ice lost, rather than the resulting sea level rise?

Line 65, "we use the Pliocene as an idealized test case to investigate how large changes in the East AIS (EAIS) height affect the climate.": This is a minor tweak, but I think it would be clearer to say something like, "we use a model of the Pliocene to investigate how large, hypothetical changes in East AIS (EAIS) height would affect the climate." I suggest this change because the Pliocene itself is not the test-case; rather, the test-cases are the hypothetical scenarios which are perturbations on the Pliocene case.

Lines 123-127, "All these sensitivity experiments are hypothetical scenarios, because changes in surface albedo due to ice sheet removal have not been accounted explicitly in the present study through increasing the sea level.": My previous comment was concerned with ice sheet volume, not surface albedo. In the mid-Pliocene warm period, the climate had time to adjust to near-modern levels of CO2. Thus, the ice sheet volume in the PRISM4 reconstruction is meant to represent a longer-term adjustment than we have thus far experienced in the present (as you mention in your introduction). The 0%, 25%, 50%, and 75% scenarios therefore represent a somewhat arbitrary further reduction *against* mid-Pliocene ice sheet volume. Would these ice sheet *volumes* correspond to any projected future scenarios, and if so which scenarios? If you make clear that these experiments are hypothetical, I don't think you necessarily need to discuss surface albedo here, since you already mention it elsewhere.

Line 136-138, "The results are presented as anomalies from the control for the sensitivity experiments, thereby estimating the EAIS height effect during the mid-Pliocene warm period.": I suggest you remove "thereby estimating the EAIS height effect during the mid-Pliocene warm period." Again, unless I'm missing some important information, the 0%, 25%, 50%, and 75% scenarios were not Pliocene scenarios. Your *control* in this study is the mid-Pliocene warm period, and the anomaly plots you show are hypothetical effects.

---

## Referee Report (RR3)

Referee Comment

**General comments**

The authors responded sufficiently to my prior comments. The current version of the manuscript presents the results more clearly and corrects the ambiguities of the original.

The authors' discussion focuses on the most apparent large-scale effects of changes in ice sheet height: the lapse-rate induced temperature feedback, the direct changes caused by redistribution of airmass over Antarctica and the rest of the globe, and the changes in moisture advection permitted by the reduced height of the Antarctic ice sheet. There are regional heterogeneities and nonlinearities in the results that warrant additional physical explanation. It would be interesting to see this, and these effects could (perhaps should) be the subject of further inquiry.

However, I think that the authors have done a good job presenting the results within the current scope of the paper, which describes the more straightforward effects of these experiments. The quality of presentation is significantly improved upon the initial submission.

---

## Author Response (AR2)

Dear Prof. Phipps,

We are submitting our revised manuscript entitled "How changing the height of the Antarctic ice sheet affects global climate: A mid-Pliocene case study" by *Huang et al.* to you for consideration for publication in *Climate of the Past*.

We are very grateful to the reviewers for the thoughtful suggestions, which have been incorporated into the revised paper (track changes). They are detailed below.

**Comments of Reviewer 1:**

Huang et al. addressed most of my comments well. However, a few of the comments were not sufficiently addressed, and there are certain points where the text remains confusing or logically inconsistent.

I think this paper could be published with minor revisions to address these issues.

**Specific comments**

Line 64, "In this scenario, Antarctica's melting ice sheets would raise sea level 20 meters in coming centuries (Grant et al., 2019).": This is not necessarily inaccurate, but it's confusing given that the aspect of AIS reduction of concern in this study is not the sea level rise but the change in the volume of the ice sheet. As you describe, could you make a statement here about the volume of ice lost, rather than the resulting sea level rise?

Done. We have changed the magnitude of sea level rise to the volume of ice lost, following the good suggestion (see lines 65-66).

Line 65, "we use the Pliocene as an idealized test case to investigate how large changes in the East AIS (EAIS) height affect the climate.": This is a minor tweak, but I think it would be clearer to say something like, "we use a model of the Pliocene to investigate how large, hypothetical changes in East AIS (EAIS) height would affect the climate." I suggest this change because the Pliocene itself is not the test-case;

rather, the test-cases are the hypothetical scenarios which are perturbations on the Pliocene case.

Thanks. We have improved the sentence following the suggestion (see lines 67-70).

Lines 123-127, "All these sensitivity experiments are hypothetical scenarios, because changes in surface albedo due to ice sheet removal have not been accounted explicitly in the present study through increasing the sea level.": My previous comment was concerned with ice sheet volume, not surface albedo. In the mid-Pliocene warm period, the climate had time to adjust to near modern levels of $CO_2$. Thus, the ice sheet volume in the PRISM4 reconstruction is meant to represent a longer-term adjustment than we have thus far experienced in the present (as you mention in your introduction). The 0%, 25%, 50%, and 75% scenarios therefore represent a somewhat arbitrary further reduction against mid-Pliocene ice sheet volume. Would these ice sheet volumes correspond to any projected future scenarios, and if so which scenarios? If you make clear that these experiments are hypothetical, I don't think you necessarily need to discuss surface albedo here, since you already mention it elsewhere.

Yes, our experiments are hypothetical. We have deleted the discussion on the surface albedo (see lines 126-132).

Line 136-138, "The results are presented as anomalies from the control for the sensitivity experiments, thereby estimating the EAIS height effect during the mid-Pliocene warm period.": I suggest you remove "thereby estimating the EAIS height effect during the mid-Pliocene warm period." Again, unless I'm missing some important information, the 0%, 25%, 50%, and 75% scenarios were not Pliocene scenarios. Your control in this study is the mid-Pliocene warm period, and the anomaly plots you show are hypothetical effects.

Many thanks for the suggestion. We have deleted the sentence "thereby estimating the EAIS height effect during the mid-Pliocene warm period" (see lines 147-148).

**Comments of Reviewer 2:**

The authors have made several useful adjustments and additions to the manuscript to address some of the issues pointed out. In contrast to what was suggested by the editor, the revisions are mostly minor and hardly any additional analysis and/or discussion was provided where requested. Therefore, a solid basis explaining the relevance of the study is still missing, as well as the elements needed for a proper physical understanding of the results shown.

In their responses, the authors seem to evade most of the issues raised and simply repeat what was already in the text, or state some trivial explanations that are often not related to the questions asked.

Many of the results and discussion focus on the obvious result, which is a lapse-rate induced temperature change and global temperature/pressure/precipitation response by redistribution of mass. The more interesting and far less intuitive responses beyond this first order effect are in my opinion left mostly untouched. I feel a more substantial effort can and should be made before publication.

**Main comments:**

L112: you mention specifically that you use dynamic vegetation here, yet you answer that all the boundary conditions are the same except for the EAIS height. I assume between your simulations? This still means that there is a difference between your MPcontrol and the original PlioMIP simulation. Please clarify.

Yes, there is a difference between our MPcontrol and the original PlioMIP simulation, namely the MPcontrol uses dynamic vegetation while the original PlioMIP simulation uses fixed vegetation. Our sensitivity experiments only changed the height of the EAIS on the basis of the MPcontrol experiment. All the results showed in the manuscript are anomalies between the sensitivity experiments and the

MPcontrol experiment, rather than those between the sensitivity experiments and the original PlioMIP simulation. The effect of the dynamic vegetation can be assessed by comparing the results between the MPcontrol experiment and the original PlioMIP simulation, and is certainly interesting, but should be a focus of another paper. To make it clearer, we have improved some sentences in the Methods section (see lines 115-120).

Temperature and precipitation responses outside of the EAIS are clearly not all linear between the different experiments. Are the linear responses you mention globally averaged, or over the EAIS only? Especially with the larger reductions, both temperature and precipitation patterns become interesting and are likely related to circulation changes in both the atmosphere and ocean. These patterns as well as their dependency to the EAIS height are not fully explained.

Yes, temperature and precipitation responses outside of the EAIS are clearly not all linear between the different experiments. In our paper, we claimed linear temperature and precipitation responses in regions over the EAIS. We actually stated "Compared with the MPControl experiment, the East Antarctic annual mean surface temperature increases by about 5 ℃, 10 ℃, 15 ℃, and 18 ℃ with the height reduction of 25%, 50%, 75%, and 100%, respectively (Figure 2 in the manuscript)". These linear responses are also confirmed by the change in temperature with height (Figure S1 in supplementary material) and the energy balance plots (Figure 8 in the manuscript). The former is similar in all of the experiments. So, at a given location temperature changes quasi-linearly with height. The later show that all factors (topography, heat transport etc.) make up a similar fraction of total temperature change in all the figures.

We agree that the temperature and precipitation changes are likely related to circulation changes in both the atmosphere and ocean. For this, we added the changes of the annual water vapor flux over Antarctica (Figure 7 in the manuscript). The results show that with the height reduction of the EAIS, the easterly flow encircle the East Antarctic continent, extending from ~60°S to the continental periphery. This

means that the water vapor flux decreases over that the continental periphery upon successive reduction of the EAIS height, which indicates that the circulation in the atmosphere and ocean are both weakened.

Improvements made to the figures are very minor and most of the issues regarding readability as well as relevance to the results and conclusions remain.

The figures in the manuscript have been further improved based on this and previous comments. First, the changes of the EAIS between the specific experiments have been added (see Figures 1b-d in the manuscript). Second, the projection and latitudinal extent of Figure 6 (Figure 7 in previous version) have been changed, in order to make it consistent with those of Figures 2 and 5 (see Figure 6 in the manuscript). Third, the energy balance between other sensitivity experiments and MPControl has been added into Figure 8 (Figure 10 in previous version) to make the explain more logic. In addition, we added one more Figure (Figure 7 in the manuscript) to make the results and conclusions clearer and another Figure to the supplementary material (Figure S1) to support the quantitative calculation.

Regardless of the changes made, section 4 is a mix of discussion and model results making this part confusing and messy. None of the analyses are presented in the methods section, and new results now seem to appear out of the blue past the main results section. The minor additions made to the methods section currently fail to resolve the overall unclear structure.

Many thanks for the suggestion. We moved the experimental design of the new sensitivity experiment, which is similar to the -100%EAIS experiment, except artificially raising the sea level by reducing the land level (away from Antarctica) by 60m, from the Discussion to Method section (lines 133-138). Moreover, the "4.4 Energy balance" section was merged into the "4.2 Causes of global temperature changes" section, to make section 4 more structured (lines 248-295).

**Specific comments:**

Correlation does not imply causation. Indeed, the redistribution of air causes global changes in pressure when changing the height of the EAIS. Using the ideal gas law, one can argue that warmer air will increase surface pressure. In reality, thermal heat lows would claim the opposite relation, as the atmospheric circulation also responds to the density anomalies. This is probably just a poor example in comparison to the study, but explaining the temperature response purely from the ideal gas law at least needs some more explaining. You could at least check whether the temperature and pressure anomalies and their spatial patterns are consistent with your hypothesis.

Over Antarctica, we show that heat transport is the main cause for the temperature changes over this region. To make it clearer, the "4.4 Energy balance" section was merged into the "4.2 Causes of global temperature changes" section (see lines 248-295). Moreover, we added the Energy balance Figure of other experiments into Figure 8 (see Figures 8b-d) and calculated the contribution of each energy balance component to the temperature changes over the East Antarctica and the rest of globe. The results show that heat transport is the primary factor and Topography (the ideal gas law) is the secondary factor for the temperature changes over Antarctica, while over the rest of the globe, Topography (the ideal gas law) is the primary factor and heat transport is the secondary factor for the temperature changes. These results have been added to the "4.2 Causes of global temperature changes" section (see lines 258-262, 264-266)

In fact, the spatial pattern of temperature and pressure anomalies have already been presented in our paper (Figures 3 and 9 in the manuscript). Evidently, the surface air pressure increases over Antarctica and decreases over elsewhere, which is similar to the spatial pattern of the air temperature changes.

Anyway, we replace "well explains" with "may explain" to tone down the argument (line 289).

The lapse rates found as a result of changing the EAIS height should be explained

better in the text as well. I am also not sure whether this calculation is correct; in the 50% reduction case there is a >18C temperature increase over the highest region of the ice sheet, which is reduced by about 2km in elevation. This would correspond to a 9C/km lapse rate rather than 5C/km. This may be completely different from how the lapse rates were calculated, but it is impossible to tell without a proper explanation.

It is true that the lapse rates are higher over the summit regions than other areas of the EAIS. However, the lapse rate we presented is the average value over the EAIS area, not for the summit area of the EAIS. To make it clearer, we calculated the lapse rates and added the Figure to the supplementary material (Figure S1) and the detailed description also has been added to the manuscript (lines 154-160).

The text still mentions that 'precipitation changes are consistent with decreased temperatures', without explaining any of the regional patterns, inconsistencies or non-linear responses. The -100% precipitation anomaly clearly is not twice the -50% one, for example, this is neither mentioned, nor explained. Another example is a slight precipitation increase over West Antarctica, where we see strong cooling. It is also unclear to me how the 5% precipitation increase per degree C was obtained, is this global average precipitation vs temperature?

We aim to address the changes over the East Antarctica versus the rest of the globe. Therefore, the regional patterns, inconsistencies or non-linear responses are not the focus of our paper, and should be topics of other papers. To make it clearer, we have improved some sentences in the introduction/abstract (see lines 23, 79).

The precipitation enhancement vs temperature is estimated only for the East Antarctica, rather than global average. Our results clearly show that the -100% precipitation anomaly clearly is twice the -50% one for East Antarctica, but not for the rest of the globe. It was obtained from the average precipitation increase dividing by the average temperature increase (see lines 184-187).

We do not agree that there is a slight precipitation increase over West Antarctica with a strong cooling. Figures 2 and 4 in the manuscript clearly show that, with the successive reduction of the EAIS height, the precipitation and temperature both

decrease over West Antarctica, in terms of the spatial resolution of the data.

I am still missing any mechanical explanation as to why the ITCZ/SPCZ would respond to EAIS changes. A thermal imbalance between hemispheres can indeed be a cause (but should then be quantified) of an ITCZ shift, but it is still unclear why the effect ramps up beyond the -50% reduction.

      The ITCZ/SPCZ is very sensitive to convective mixing, which is closely related to the intensity of trade winds. As shown in Figure 3 in the manuscript, beyond the -50% reduction, the meridional gradient of surface air temperature decreases over the low to mid-latitude Western Pacific, leading to a weakening of the trade winds and decreased water vapor flux (Figure 1 below). This may explain the decreased precipitation over most of the tropical areas.

[Figure]

Figure 1. Vertically integrated water vapor flux (arrows, units: kg m$^{-1}$ s$^{-1}$) for (a) the MPControl, and the anomalies (arrows, units: kg m$^{-1}$ s$^{-1}$) for (b) the -100%EAIS relative to the MPControl, (c) the -75%EAIS relative to the MPControl, (d) the -50%EAIS relative to the MPControl, and (e) the -25%EAIS relative to the MPControl.

      The Zonally averaged temperature changes (Figure 2 below) show that the temperature increases significantly over southern hemisphere while changes very slightly over northern hemisphere, indicating a clearly thermal imbalance between hemispheres. ITCZ shift may play an additional role, but factors influencing it

remains unclear, including changes in the balance of heat between the hemispheres, the wind-driven ocean circulation, and convective mixing and so on (Donohoe et al. 2014; Adam et al. 2016; Green and Marshall, 2017; Talib et al., 2020), which requires systematic investigation in future studies.

[Figure]

Figure 2. Zonally averaged temperature changes (units: °C) for (a) the -100%EAIS relative to the MPControl, (b) the -75%EAIS relative to the MPControl, (c) the -50%EAIS relative to the MPControl, and (d) the -25%EAIS relative to the MPControl.

It is pretty much impossible to see the change in katabatic winds from figure 7. Although it may be straightforward, the wind field alone is not enough to explain moisture transports without knowing the actual moisture field. Even in the current climate, katabatic winds are confined to the area very near the ice sheet's edge, making it tough to explain moisture transports over a large latitudinal range from this effect only.

We have added water vapor flux figures to further support the explanation (Figure 7 in the manuscript and lines 235-247).

The statement that responses to the EAIS are linear are still not substantiated by any clear figure or clear quantitative assessment. Without such, it is a claim that cannot be validated nor explained.

Do not agree. As mentioned earlier, the precipitation and temperature both

increase over East Antarctica with the successive reduction of the EAIS height, which are clearly presented in Figures 2 and 4 in the manuscript. We added the calculation in the text (lines 156-160, 184-187). Moreover, these linear responses are also confirmed by the change in temperature with height (Figure S1 in supplementary material) and the energy balance plots (Figure 8 in the manuscript). The former is similar in all of the experiments. So, at a given location temperature changes quasi-linearly with height. The later show that all factors (topography, heat transport etc.) make up a similar fraction of total temperature change in all the figures.

If the study focuses only on the effects over Antarctica versus the rest of the globe, this is currently unclear from the abstract/introduction. As you show in the energy balance analysis, heat transports are the primary contribution to much of the changes. Yet, you claim that the ideal gas law and temperature changes are mostly responsible. This is at least partly contradictory, as the heat transport suggest that circulation changes should have at least a comparable contribution. While mentioning this contradiction yourself, the ideal gas law explanation is still presented as the main mechanism in section 4.2.

Yes, we aim to address the effects over the East Antarctica versus the rest of the globe. In fact, we have already mentioned this in the introduction, which is further clarified in both the introduction (line 79) and abstract (line 23) in the revised manuscript.

Our results show that over the Antarctica, heat transport is the primary factor influencing temperature, and the topography (which represents the ideal gas law) and GHG play a secondary role (turquoise line in Figure 8 in the manuscript), while over the rest of globe, the topography (which represents the ideal gas law) and GHG are the primary factor influencing temperature, and heat transport plays a secondary role. In Section 4.2, we did not say that the ideal gas law is the 'primary' factor. To make the expression clear and logical, the "4.4 Energy balance" section was improved and merged into Section 4.2 (lines 248-295).

The experiment in which the land surface is decreased by 60m does not act to support the direct link between temperature and pressure. It merely shows that the mass loss of the AIS that was previously unaccounted for does not substantially alter the results. If anything, Figure 9 shows that outside of the AIS, where lapse rate effects dominate, hardly any spatial correlation remains between the temperature and pressure responses.

The experiment (-60 m land surface) is not designed to act to support the direct link between temperature and pressure. In fact, this experiment is designed to provide a more realistic -100%EAIS experiment by lowering the land surface (artificially raising the sea level), and acts to verify the cooling effect of EAIS height change over extra-Antarctic regions.

Our paper focus on the comparison of the EAIS height effect between East Antarctica and the rest of the globe. The temperature contrast between them is well explained by heat transport and surface air pressure change, as evidenced by the energy balance results (lines 248-295). The temperature and pressure anomalies are much smaller over extra-Antarctic regions compared to those over Antarctica, which may result from complex interactions between ocean and atmosphere. This requires further study.

**Figures:**

Figure 1: this is a nice addition, but does not show any new information compared to what can be found in previous PlioMIP publications. The aim of such a figure would be to show how the EAIS was changed between the specific experiments.

Done. The changes of the EAIS between the specific experiments have been added (see Figures 1b-d in the manuscript).

Figures 2-4: as figures 2 and 3 show the same field over a different region, while figure 4 shows a very similar field over the same region, at least one of them is redundant in the current set-up. Of course, SAT and SST effects are closely related

over the ocean, as they are both at or near the surface and therefore nearly the same.

Done. The SST effects have been moved to supplementary materials.

Figure 7: The projection and latitudinal extent used here is not at all consistent with Figures 2 and 5, so I fail to see what kind of consistency is meant here. Regardless of consistency, the figure remains near impossible to read and interpret. Cylindrical versus stereographic projection will not make much of a difference when looking only at the pole, but the former becomes very unrealistic when showing an entire hemisphere.

Thanks. We have changed the projection and latitudinal extent of Figure 6 in the manuscript (Figure 7 in previous version), in order to make them consistent with those of Figures 2 and 4 (Figure 5 in previous version).

**References**

Adam, O., Bischoff, T., and Schneider, T.: Seasonal and interannual variations of the energy flux equator and ITCZ. Part I: Zonally averaged ITCZ position, J. Climate, 29, 3219–3230, doi:10.1175/JCLI-D-15-0512.1, 2016.

Donohoe, A., Marshall, J., Ferreira, D., Armour, K., and McGee, D.: The interannual variability of tropical precipitation and interhemispheric energy transport. J. Climate, 27, 3377–3392, doi:10.1175/JCLI-D-13-00499.1, 2014.

Green, B., and Marshall, J.: Coupling of trade winds with ocean circulation damps ITCZ shifts. J. Climate, 30(12), 4395–4411, 2017.

Talib, J., Woolnough, S. J., Klingaman, N. P., and Holloway, C. E.: The effect of atmosphere-ocean coupling on the sensitivity of the ITCZ to convective mixing. J. Adv. Model Earth Sy., 12, e2020MS002322, doi:10.1029/2020MS002322, 2020.